# Leveraging Attention-Based Convolutional Neural Networks for Meningioma Classification in Computational Histopathology

**DOI:** 10.3390/cancers15215190

**Published:** 2023-10-28

**Authors:** Jannik Sehring, Hildegard Dohmen, Carmen Selignow, Kai Schmid, Stefan Grau, Marco Stein, Eberhard Uhl, Anirban Mukhopadhyay, Attila Németh, Daniel Amsel, Till Acker

**Affiliations:** 1Institute of Neuropathology, Justus Liebig University Giessen, Arndtstr. 16, D-35392 Giessen, Germany; jannik.sehring@patho.med.uni-giessen.de (J.S.);; 2Department of Neurosurgery, Hospital Fulda, Pacelliallee 4, D-36043 Fulda, Germany; 3Department of Neurosurgery, University Hospital Gießen, Klinikstr. 33, D-35392 Giessen, Germany; 4Department of Computer Science, Technical University of Darmstadt, Fraunhoferstraße 5, D-64283 Darmstadt, Germany

**Keywords:** computational histopathology, deep neural networks, DNA methylome, meningioma, multiple-instance learning

## Abstract

**Simple Summary:**

Meningioma is the most common primary intracranial tumor. DNA methylation-based subtyping, while highly useful for diagnosis and treatment planning, is costly and not widely available. Therefore, the identification of methylation classes based on histological sections would be highly beneficial as it could greatly support and accelerate diagnostic and treatment decisions. We developed and systematically evaluated an AI framework to perform the classification of the most prevalent methylation subtypes based on histological sections. The model achieved a balanced accuracy of 0.870 for benign-1 vs benign-2 and 0.749 for benign-1 vs. intermediate-A in a narrow validation set. Combined with the network’s assessed focus on key tumor regions these results provide a promising proof-of-concept of such an AI-driven classification approach in precision medicine.

**Abstract:**

Convolutional neural networks (CNNs) are becoming increasingly valuable tools for advanced computational histopathology, promoting precision medicine through exceptional visual decoding abilities. Meningiomas, the most prevalent primary intracranial tumors, necessitate accurate grading and classification for informed clinical decision-making. Recently, DNA methylation-based molecular classification of meningiomas has proven to be more effective in predicting tumor recurrence than traditional histopathological methods. However, DNA methylation profiling is expensive, labor-intensive, and not widely accessible. Consequently, a digital histology-based prediction of DNA methylation classes would be advantageous, complementing molecular classification. In this study, we developed and rigorously assessed an attention-based multiple-instance deep neural network for predicting meningioma methylation classes using tumor methylome data from 142 (+51) patients and corresponding hematoxylin-eosin-stained histological sections. Pairwise analysis of sample cohorts from three meningioma methylation classes demonstrated high accuracy in two combinations. The performance of our approach was validated using an independent set of 51 meningioma patient samples. Importantly, attention map visualization revealed that the algorithm primarily focuses on tumor regions deemed significant by neuropathologists, offering insights into the decision-making process of the CNN. Our findings highlight the capacity of CNNs to effectively harness phenotypic information from histological sections through computerized images for precision medicine. Notably, this study is the first demonstration of predicting clinically relevant DNA methylome information using computer vision applied to standard histopathology. The introduced AI framework holds great potential in supporting, augmenting, and expediting meningioma classification in the future.

## 1. Introduction

Medical images and their accompanying metadata contain a wealth of shared and complementary information. By leveraging computer vision techniques, the shared subspace between these modalities can be harnessed, while complementary information can contribute to the improvement of precision medicine through the integration of different information modalities. In parallel, computational pathology has gained prominence in the diagnosis of oncological diseases in recent years, primarily due to advancements in artificial intelligence (AI) [1,2,3,4,5]. The application of machine- and deep-learning methods to digitized histopathology images promises unprecedented accuracy, efficiency, and reproducibility in precision medicine, laying the foundation for the implementation of computer-aided decision support systems in clinical practice [1,2,3,4,5,6,7].

In pioneering studies, deep-learning approaches have been applied to aid histological diagnosis or predict molecular information from standard HE-stained tissue slides. For instance, a weakly supervised deep-learning algorithm developed by Campanella et al. [1] was able to recognize areas of tumor tissue in prostate cancer, basal cell carcinoma, breast cancer metastases, and lymph nodes with high accuracy. A clinical decision support system was developed that selects only relevant slides from each case for review by a pathologist. Recent studies reported deep-learning algorithms capable of predicting the origin of multiple cancer types of unknown primary [2] or classifying histological subtypes in renal cell carcinoma and lung cancer [3]. Recent AI-driven approaches have successfully linked digital histopathological and molecular data to identify various genetic alterations or predict molecularly defined tumor subtypes for some tumor types, including lung, colon, breast, and gastric cancer [8,9,10,11,12,13,14,15,16,17,18,19]. These findings suggest that deep convolutional networks can be employed to predict molecularly defined and prognostically relevant cancer subtypes and gene mutations from computerized histological images [14,20,21,22,23]. Meningiomas, the most common primary intracranial tumors, originate from the lining of the brain and spinal cord [24,25]. The current WHO CNS5 classification designates meningioma as a single tumor type with 15 histological subtypes, 9 of which correspond to WHO grade 1 and three, each to CNS WHO grade 2 and WHO grade 3 as defined by histology. However, classification and grading criteria are continuously being updated and refined due to the growing knowledge gained through the implementation of molecular markers [26]. Despite some initial studies in gliomas [11,12,15,27,28,29], the application of computational histopathology in neuro-oncology for predicting molecular tumor subtypes remains largely unexplored, particularly in meningioma research. This gap can be partly attributed to the limited availability or absence of public datasets containing matched images and molecular data for this tumor type. For instance, matched DNA methylome and histopathology data from meningiomas are not available in TCGA, the public database most commonly used for this type of study. However, recent efforts to support the differential diagnosis of meningioma by digital histopathology suggest that whole-slide-images (WSI) (without AI-assisted analysis but evaluated by neuropathologists) are similarly useful as traditional microscopy slides for diagnosing atypical meningiomas [30] and that automated meningioma grading using WSI and AI algorithms based on mitotic count is feasible [7]. Methylome profiling has emerged as a powerful method for classifying CNS (central nervous system) tumors into relevant predictive classes, including meningiomas [31,32]. By assessing genome-wide DNA methylation patterns, a study by Sahm et al. [32] distinguishes six distinct classes of meningiomas: three benign classes (benign-1, benign-2, benign-3), two intermediate classes (intermediate-A, intermediate-B) and one malignant class. These represent clinically and biologically more homogeneous groups than the 15 histologically defined subtypes and allow for more precise segregation of patients with low to high risk of progression [33]. DNA methylation profiling, while highly useful for tumor classification, is costly, time-consuming, and requires specialized equipment that is not widely available. Consequently, predicting methylation pattern-derived tumor classes from routinely examined histological data with analytical performance comparable to molecular analyses holds significant clinical value. Of importance to note is that machine learning algorithms have demonstrated their capacity to correlate morphometric features of WSIs with DNA methylome profiles in glioma and renal cell carcinoma [29], demonstrating that the common subspace of the two modalities can be exploited with AI-assisted approaches in tumor classification.

Here, we present the results of a proof-of-principle investigation, using whole-slide-images (WSI) and an attention-based multiple-instance learning approach, to predict methylation classes in meningiomas by employing retrospective in-house datasets. We applied AI-assisted morphological analysis of standard hematoxylin-eosin-stained (HE) tissue sections to distinguish the most prevalent methylation classes of meningiomas (benign-1, benign-2, and intermediate-A) [32]. Notably, pairwise analyses revealed highly accurate classification within two combinations of meningioma methylation classes using the deep-learning approach. Our work highlights the potential utility and current limitations of AI-assisted morphomolecular analysis classification of meningioma. We believe that this study serves as an initiative to incorporate AI-assisted digital histopathology into the existing repertoire of diagnostic and prognostic methods for meningiomas and potentially other CNS tumors.

## 2. Materials and Methods

### 2.1. Tumor Samples, Methylome Datasets, and Classification

Tumor samples of patients with confirmed histological diagnosis of meningioma according to the WHO 2021 grading scheme [26] were used for DNA methylation analysis and generation of a WSI dataset (see Section 2.2). DNA was extracted from representative tumor areas of interest, as highlighted by a neuropathologist, and analyzed with an Illumina HumanMethylation Infinium EPIC BeadChip (850k) array as specified and described by the manufacturer. Methylome data were classified with the Meningioma Classifier [31,32] to retrieve the corresponding meningioma methylation class and the corresponding probability score. All samples of this study were collected and analyzed at the Giessen Institute of Neuropathology in a pseudonymized manner as approved by the institutional review board.

### 2.2. Whole-Slide Image Datasets

Standard hematoxylin-eosin staining was performed on 3–4 µm thin formalin-fixed, paraffin-embedded (FFPE) tissue sections using the Ventana benchmark Ultra (Roche, Basel, Switzerland) automated stainer. The glass slides were digitized using a Hamamatsu Nanozoomer S360 (Hamamatsu, Japan) at 40× equivalent magnification resulting in WSIs with a resolution of 0.23 µm/px. As predictions are intractable at the WSI level due to memory restrictions, a bag of smaller patches was generated from the WSIs, which are called tiles. Each tile has a size of 256 × 256 pixels, without overlap between the tiles. This size was chosen according to the encoder network and the available memory. Tiles were extracted at native 40× equivalent magnification and from downsampled 20× equivalent magnification of the WSI for each patient.

### 2.3. Dataset Curation

Only patients with available WSI and methylation classification data were considered, and with a methylation classification probability score above 0.5, according to the Meningioma Classifier [31,32], were used for further analysis. We divided the patients into two distinct subsets. The larger subset was used for development (development sets: A, C). The second subset was considered a narrow validation set according to Kleppe et al. [34] as it is not used during the development of the neural network (narrow validation sets: B, D). We further used the availability of additional HE slides per patient on which areas of interest (ca. 0.7–69 mm${}^{2}$) corresponding to representative tumor sections that were highlighted by an experienced neuropathologist (datasets C, D). Areas of interest were marked with a pencil by clinical professionals during routine diagnostic work prior to the initiation of our study and were thus performed in a manner blinded to our experiments, preserving the integrity and impartiality of the process. We subsequently labeled those areas with QuPath [35] on the WSI. DNA used for the DNA methylation array and classification was extracted from these areas. Accordingly, the labeled areas analyzed using AI-assisted digital histopathology contained the tumor sites that were subjected to molecular analysis. This allows for additional evaluations regarding tumor heterogeneity (see Section 3.3) and focus of the network on areas of interest (see Section 3.4). Consequently, we ended up with 4 datasets: Dataset A and C for development, where C contains labeled slides, and Dataset B and D for narrow validation, where D contains labeled slides. Any differences between datasets A and C, as well as B and D, were due to the unavailability of clearly marked slides. Due to the very different number of tumor cases in the different classes in our real-life clinical setting (Table 1) and the necessity for a large group size for AI-based algorithms, we selected the largest methylation classes, namely benign-1 (n = 47), benign-2 (n = 71) and intermediate-A (n = 75), for further analysis. This resulted in three two-class classification problems namely benign-1 vs. benign-2, benign-1 vs. intermediate-A, and benign-2 vs. intermediate-A.

Importantly, all three classes belong to methylation Group A, along with the rarely diagnosed methylation class benign-3. Group B consists of the methylation classes intermediate-B and the malignant class. Thus, an analysis of specimens from the methylation classes benign-1, benign-2, and intermediate-A provides an ideal setup to initially test the analytical performance of AI-assisted morphological diagnosis for the following reasons: (i) the number of samples per methylation class allows for an assessment of patient diversity, (ii) the cohort sizes are similar, and (iii) the relative similarity of methylomes (all Group A) coupled with differences in predominant histological and genetic features allows for a critical assessment of resolution performance. The number of patient samples in each dataset is shown in Table 1. Additionally, WHO grades based on histological and molecular features are given to show the discrepancies between the WHO grading approach and the methylation classification scheme. Each of these approaches classifies tumors into distinct grades. Due to the non-overlapping groups, additional information on tumor grade can be obtained independently from both classification schemes.

### 2.4. Image Data Preprocessing

As a preprocessing step, the background was filtered out via Otsu’s thresholding [36] and morphological image operations at the lowest resolution of the WSIs image pyramid. Morphological image operations were performed to remove smaller fragments and fill small holes to obtain a consecutive tissue mask. For datasets C and D, everything outside the pen-labeled area was additionally considered as background. Additional filtering of blood, based on a fixed cutoff for red values, and background with the original Otsu value, as well as morphological operations, were performed again on the individual tiles. To exclude uninformative tiles, a tile was only used for further analysis if the estimated percentage of tissue was over 75% without background and blood.

### 2.5. MIL-Based Slide Diagnosis

To predict the methylation class of a patient, we utilized a two-class weakly supervised learning setting, with the WSI as input data. As it is infeasible to feed in the high-resolution image at once, we made use of a multiple-instance learning approach similar to that proposed by Lu et al. [3]. To extract features of each tile, the first layers of a convolutional neural network, a ResNet50 [37], pre-trained on ImageNet, were used. This resulted in a feature vector with a dimensionality of 1024 per tile. ResNet50 was chosen as it is a well-established baseline for image recognition. This step was necessary to reduce the dimensionality of the input further. The combination of tiles extracted at 20× and 40× allowed us to aggregate high-resolution local information as well as more global information retrieved at a lower resolution and, therefore, incorporate possible larger structures as features. Smaller magnifications were not considered as they may hold too much background per tile and may not hold significant additional information. As feature vectors were only generated once before training the attention network, no augmentation was performed on the tile level besides image normalization as needed for ResNet50. In our study, every specimen was prepared, sectioned, stained, and digitized in the same manner within our institute. This meticulous consistency rendered color augmentation unnecessary. Instead, we used dropout to prevent the neural network from overfitting and to regularize the model training. In the field of augmentation, it is common to use spatial in addition to color augmentation. Here, we distinguish between elastic deformations and rotations or reflections. We have deliberately forgone elastic augmentation to preserve and not artificially change the characteristic cell shapes, which could compromise and possibly generate data that does not exist in nature. By default, CNNs are not rotation invariant by design. Yet, with our dataset encompassing over 7 million tiles with up to 100,000 tiles per WSI, incorporating rotations and mirroring would have been computationally exhaustive and considerably lengthened the training process. as the feature vectors would be encoded in every epoch. Similarly, Lu et al. [3] also waived this step in their model generation. Another way to perform augmentation could be by augmenting the bag of feature vectors fed into the attention network. To obtain different bags per patient, we decided to randomly discard up to 25% of the feature vectors per patient in the training phase. It turned out that this had no positive effect when testing with five Monte Carlo runs for each classification task. Therefore, we decided not to include this type of augmentation in our experiments. Next, all concatenated feature vectors per patient were fed into a multi-branch attention network with fully connected layers and a gated attention mechanism. This multi-branch attention network first shares a common branch across all possible classes and afterward splits it into one branch per class, thus allowing the use of information shared across classes, as well as learning class-specific features. This network learns an attention value per feature vector, and weighted feature vectors are aggregated patient-wise with global average pooling. This resulted in a probability for each class per patient normalized using the SoftMax activation function. The development dataset was randomly divided at the patient level into training (70%), validation (10%), and test (20%) sets utilizing Monte Carlo cross-validation. Testing during the development phase allows us to properly assess the generalizability of the network while the narrow group remains untouched, and these patients have no influence on the model development. For splitting, each methylation class was considered independently to keep the original ratio of the classes. For the final classification, the maximum class prediction score was taken to avoid error-prone threshold tuning over those prediction scores. Details on the training of the network (Section 7) and the hardware and software used (Section 6) can be found in the Appendix A.

### 2.6. Visualization of Attention Values

Attention values per tile were stored and visualized using QuPath [35] to enable a manual and interactive inspection of the relative importance of the individual tiles for the prediction. For this, attention values were ranked according to the values in the attention map of the predicted class. We assigned a color value to each of the resulting ranks according to a red-to-blue continuous color gradient map. Red indicates a tile of high relative importance, while blue indicates low relative importance (Figure 1). For visualization purposes, 20× and 40× maps can also be considered independently. Ranks are then only computed within one map to get a higher resolution within the predefined color spectrum at one magnification.

### 2.7. Compatibility with Current Reporting Standards

The documentation of this work is compliant with the recommendations and guidelines for reporting minimum information about clinical artificial intelligence modeling (MI-CLAIM) as proposed by Norgeot et al. [38]. Information on sex, age, methylation classification, and WHO grade of each sample and their assignments in datasets A–D, as well as a report with minimum information on clinical artificial intelligence modeling (MI-CLAIM), according to Norgeot et al., are available on request.

### 2.8. Informed Consent and Ethics Approval

Informed consent was obtained from all participating subjects, and the experimental studies were authorized by the ethics committee of Justus Liebig University Giessen (AZ138/18 and 07/09).

## 3. Results

### 3.1. AI-Assisted Determination of Meningioma Methylation Classes in Two-Class Setups from WSI Data

We assessed the performance of the AI-assisted morphological classification of methylation classes using balanced accuracy values, which also account for class imbalances present in the data distribution, thus giving more reliable results than simple accuracy values. Details on computational analyses can be found in the supplement. We conducted Monte Carlo cross-validation with 50 runs and obtained a 95% confidence interval for balanced accuracy of 0.896 ± 0.151 for the benign-1 vs. benign-2 combination and 0.830 ± 0.146 for benign-1 vs. intermediate-A. For benign-2 vs. intermediate-A, we were not able to distinguish the classes effectively, achieving a 95% confidence interval of 0.542 ± 0.176, suggesting results close to random guessing. In addition, the diagnostic performance for each pairwise classification is illustrated by computing the sensitivity vs. specificity as standard ROC curves. Mean areas under the curve (AUC) ± one standard deviation are provided for quantitative evaluation. Benign-1 vs. benign-2 (sensitivity for benign-2) with a mean AUC of 0.95 ± 0.04, and benign-1 vs. intermediate-A (sensitivity for intermediate-A) with a mean AUC of 0.90 ± 0.07 confirmed the high accuracy of our approach in discriminating benign-1 from either benign-2 or intermediate-A WSIs. In agreement with the low balanced accuracy value, benign-2 vs. intermediate-A classification had a mean AUC of 0.58 ± 0.11, indicating unsuccessful discrimination in our experiments.

### 3.2. External Cohort Evaluation of the AI-Assisted Meningioma Classification

To further evaluate the diagnostic performance of our approach, we analyzed a second set of in-house patient data (n = 51, analyzed dataset B) that was not included in the previous dataset (Dataset A) used for training the deep-learning models. This experimental setup is considered a *narrow validation* according to Kleppe et al. [34] The cohort in Dataset B included benign-1 (n = 13), benign-2 (n = 19), and intermediate-A (n = 19) methylation class meningioma samples. We conducted testing on Dataset B using n = 50 runs created with the Monte Carlo cross-validation. Balanced accuracy values and ROC curves were calculated as before for Dataset A to assess the diagnostic performance and sensitivity of one-class detection per two-class classification. The results from the external Dataset B analyses were in good agreement and strongly supported those obtained from the development Dataset A. Therefore, for the benign-1 vs. benign-2 combination, we achieved a 95% confidence interval for balanced accuracy of 0.870 ± 0.067, and for benign-1 vs. intermediate-A 0.749 ± 0.058. For benign-2 vs. intermediate-A, results were close to random guessing with an interval of 0.593 ± 0.124 (Figure 2a). Consistent with these findings, ROC curves showed a mean AUC of 0.93 ± 0.01 for benign-1 vs. benign-2 (sensitivity for benign-2), mean AUC of 0.81 ± 0.02 for benign-1 vs. intermediate-A (sensitivity for intermediate-A) and mean AUC of 0.66 ± 0.07 for benign-2 vs. intermediate-A (sensitivity for intermediate-A) (Figure 2b–d).

### 3.3. Network Training on Restricted Area Dataset

WSIs often contain diverse non-tumor tissues such as hemorrhage, bone, brain tissue, or tissue artifacts that might compromise the diagnostic accuracy of the AI-assisted classification. In addition, tumor heterogeneity may also be associated with differences in the DNA methylome that are not fully captured and represented in the methylome profiling data. We, therefore, tested whether training on preselected tumor areas with available methylome data from the same region could consequently improve network performance. We trained our deep-learning model with Dataset C (which contained representative tumor sections selected by an experienced neuropathologist) and evaluated the model by narrow validation with Dataset D. In advance, we trained on Dataset C without restricting the tiles to the representative tumor sections to ensure that the change to Dataset C from Dataset A as in our previous results did not lead to unwanted side effects. We received a 95% confidence interval of 0.906 ± 0.143 (compared to 0.896 ± 0.151 from Dataset A) for benign-1 vs. benign-2, 0.852 ± 0.142 (compared to 0.830 ± 0.146 from Dataset A) for benign-1 vs. intermediate-A and 0.616 ± 0.199 (compared to 0.542 ± 0.176 from Dataset A) for benign-2 vs. Intermediate-A. This corroborates the comparable performance of Dataset C to Dataset A. For the benign-1 vs. benign-2 class combination in Dataset C, we achieved a 95% confidence interval for balanced accuracy of 0.828 ± 0.220, and for benign-1 vs. intermediate-A 0.810 ± 0.200. For benign-2 vs. intermediate-A, the results were close to random guessing with a balanced accuracy of 0.532 ± 0.215. ROC curves showed a mean AUC of 0.90 ± 0.09 for benign-1 vs. benign-2 (sensitivity for benign-2), a mean AUC of 0.86 ± 0.10 for benign-1 vs. intermediate-A (sensitivity for intermediate-A), and mean AUC of 0.56 ± 0.14 for benign-2 vs. intermediate-A (sensitivity for intermediate-A). In the narrow validation Dataset D, for the benign-1 vs. benign-2 combination, we achieved a 95% confidence interval for balanced accuracy of 0.813 ± 0.069, for benign-1 vs. intermediate-A 0.751 ± 0.072. For benign-2 vs. intermediate-A, the results were again close to random guessing with a balanced accuracy of 0.589 ± 0.134 (Figure 3a). ROC curves revealed a mean AUC of 0.89 ± 0.02 for benign-1 vs. benign-2 (sensitivity for benign-2), a mean AUC of 0.86 ± 0.03 for benign-1 vs. intermediate-A (sensitivity for intermediate-A), and a mean AUC of 0.67 ± 0.06 for benign-2 vs. intermediate-A (sensitivity for intermediate-A) (Figure 3b–d). Thus, training on preselected tumor areas did not improve the performance of the deep neural network. Taken together, these results suggest that using tumor and diverse tissue areas located outside the tumor area subjected to molecular profiling does not substantially affect classification performance in AI-based histopathological analyses. Instead, the restriction to a smaller number of tiles—approximately by a factor of 10—slightly weakens the performance in the best-performing benign-1 vs. benign-2 case despite a presumably higher data quality.

### 3.4. Statistical Assessment and Visualization of Attention Maps

To gain insight into the histological tumor areas that the attention network deems relevant for proper meningioma classification, we evaluated the attention maps generated from the attention network’s prediction. As attention maps provide a view into the model’s decision-making process, we evaluated them statistically beyond visual inspection. We used models trained on Dataset A as they performed better compared to Dataset C. To assess whether the attention network focuses on structures that an experienced neuropathologist would also consider relevant, we evaluated the proportion of high-attention tiles inside and outside areas that were highlighted as highly representative, using Dataset D. As additional evaluation revealed the higher importance of 20× magnification tiles, we focused on their attention maps. We calculated the proportion of high and low attention tiles inside and outside the marked area per patient and per run, averaging over both (Figure 4 and Figure 5 and Appendix A). In both benign-1 vs. benign-2 and benign-1 vs. intermediate-A combinations, the proportion of high-attention tiles was enriched inside the marked areas. This was not the case for benign-2 vs. intermediate-A, highlighting a reciprocal relationship between attention proficiency and classification performance. Additionally, we evaluated focus robustness and relative importance of 20×/40× magnification in class determination as described in detail in the supplement (Section 10). We also included additional analyses of inside/outside attention map statistics differentiated between correctly and incorrectly classified patients for all three classification tasks (Appendix A). Taken together, the attention network demonstrated its ability to identify regions of interest that correspond with those highlighted by an experienced neuropathologist. Since the neuropathologist highlighted areas with a high and representative tumor cell content, these results indicate that the network can detect such regions and preferentially aggregate its prediction from them. This suggests that the network can focus on histological features relevant to accurate classification.

## 4. Discussion

In this study, we introduce an AI-assisted image analysis approach that leverages the visual decoding capabilities of convolutional neural networks (CNN) to identify prognostically relevant, methylome-defined tumor classes of meningiomas [32,33] using conventional HE-stained histopathology slides. Our analysis of larger sample cohorts from a real-world clinical setting demonstrates a high degree of accuracy in classifying two combinations of meningioma methylation classes (benign-1 vs. benign-2 and benign-1 vs. intermediate-A groups). These results underscore the growing importance of computational approaches for image-based predictions of clinically relevant tumor subtypes based on molecular alterations to support and assist precision medicine in oncology. To our knowledge, this is the first study to successfully extract clinically relevant DNA methylome information through computer vision techniques applied to standard histopathology.

Our primary objective was to predict methylation classes in meningiomas using a deep-learning framework based on histological features—a task not achievable at the neuropathologist level—as DNA methylation-based molecular classification has demonstrated higher predictive power for tumor recurrence than histopathological classification alone [33]. The evaluation of balanced accuracy values and ROC curves showed high accuracy in discriminating benign-1 from either benign-2 or intermediate-A WSIs. However, the benign-2 vs. intermediate-A classification yielded results close to random guessing, suggesting that these classes are not successfully discriminated in our experiments. Intriguingly, this might be attributed to the closer relationship between benign-2 and intermediate-A classes on the methylation level, as demonstrated by the unsupervised hierarchical clustering data from Sahm et al. [32], which could also be reflected in the histological tissue level explaining the poorer diagnostic performance of the CNN in distinguishing these methylation classes. Interestingly, the classification of intermediate-A class samples using an updated version of the DNA methylome classifier (V12.5) and a different subspace of the methylome also suggests a close relationship between intermediate-A and benign-2 classes. This would imply that our AI approach facilitates a more robust definition of (morphomolecular) meningioma classes than methylome analyses alone. This hypothesis should be tested in future experiments if the updated methylome-based analyses reach a consensus in defining new methylation classes.

One limitation of our study is the small cohort size for some methylation classes, which precluded multi-class comparisons. This situation reflects the real-life clinical setting, as approximately 90% of patient samples in our retrospective dataset belong to only three methylation classes (benign-1, benign-2, or intermediate-A). Consequently, the other three methylation classes (benign-3, intermediate-B, and malignant) were not included in the deep-learning framework due to their small cohort size. To enhance the applicability of our framework in a clinical setting, we are currently gathering samples for the underrepresented meningioma methylation classes to enable multi-class comparisons and incorporate additional molecular methylome-based meningioma classifications [33,39,40]. This will help determine which molecular stratification offers the most accurate translation between histological and molecular data and evaluate the potential of integrating both modalities to improve meningioma stratification in clinical settings. Overall, our study demonstrates the ability of CNNs to identify prognostically relevant methylome-defined tumor subtypes, substantially advancing previous findings on integrating DNA methylation and morphological data, e.g., in glioma and renal cell carcinoma [29], as well as in breast cancer [8]. Several multiple-instance learning approaches have been published in the past to predict molecular features from histology data. The approach presented here extends these to uniquely enable methylation-based tumor classification of meningiomas, a previously unexplored combination of molecular features and tumor type. Recent approaches tackled different molecular questions in (neuro-)pathology. Jiang et al. [12] predicted prognosis and IDH status on LGGs using a ResNet18 for feature encoding and fully connected layers for final classification. They achieved an AUC (area under the curve) of 0.667 for IDH status prediction. Liu et al. [15] also predicted the IDH status in LGGs and GBMs, using a ResNet50 for the prediction and additionally utilizing GANs (Generative Adversarial Network) for data augmentation. They achieved an accuracy of 0.853. Zheng et al. [29] identified differential methylation states in glioma and renal cell carcinoma using morphometric features and classical machine learning. They achieved an average AUC of 0.74 in glioma samples. Similar to other AI-assisted approaches for molecular stratification of histopathological samples, we also adopted a multiple-instance strategy. Notably, Cui et al. [11] utilized CNNs as well to obtain a patch-level score and aggregated those into a classification using MIL pooling. They achieved an AUC of 0.84 for IDH1 status prediction in glioma. Campanella et al. [1] classified positive and negative samples in prostate cancer, basal cell carcinoma, and breast cancer metastases, using a combination of ResNet34 for feature encoding and an RNN to aggregate the information in a multiple-instance learning manner. They achieved an AUC above 0.98 for all their classification tasks. Courtiol et al. [6] used a ResNet50 to obtain a score per tile and aggregated them using a multi-layer perceptron to predict the patient outcome in mesothelioma. They achieved a c-index of 0.643.

By showcasing the feasibility of predicting methylome-based tumor subtypes in meningiomas through computational approaches, our research contributes to the broader understanding of AI’s potential in tumor classification and diagnosis.

Interestingly, training the network on a preselected and highly representative tumor area Dataset C—at the same time excluding non-tumor tissue, tissue artifacts, and potential tumor heterogeneity, which is not captured during DNA methylome profiling—did not improve its performance. In contrast, in Dataset A, the training on a higher amount of tiles—approximately by a factor of 10—and with higher variance in tissue diversity resulted in improved performance while allowing the network to ignore irrelevant tissue sections. This finding indicates that restricting the dataset to a smaller number of tiles and a reduced variance in relevant tissue/tumor areas during training (Dataset C) may slightly reduce the performance and weaken the generalization ability of the network. This is in concordance with previous findings [3] showing that the performance of networks improves with increased sample size. The attention maps generated from the attention network’s predictions allowed us to assess which histological tumor areas the network considered relevant for proper meningioma classification. The results indicate that the network is capable of detecting regions with a high and representative tumor cell content and preferentially aggregates its prediction from these regions, aligning with the areas highlighted by an experienced neuropathologist. This finding supports the notion that AI-based histopathological analyses can effectively learn and focus on relevant histological structures, increasing the transparency and interpretability of deep-learning models in medical applications as previously reported in gliomas [9,11,12,15,27,28,29].

## 5. Conclusions

Overall, our study serves as an important proof-of-principle for integrating AI-assisted digital histopathology into existing diagnostic and prognostic methods for meningiomas and possibly other CNS tumors. We demonstrate the potential of CNNs to extract clinically relevant methylation patterns from histological features. As we continue to refine these techniques and expand their application to different molecular classifications, as discussed above, we can expand their utility and move closer to more accurate and clinically relevant meningioma stratification, ultimately improving patient care and outcomes. It will be interesting to see whether, in the future, these features can also be made apparent to the human (neuropathologist’s) eye. Our approach could expand the repertoire of existing methods for the diagnosis and prognosis of meningiomas, and possibly other tumors, in keeping with the idea put forth by Rudolf Virchow that “each anatomical change must have been preceded by a chemical one” [41].

## 6. Hardware and Software

All computations were performed on a single Linux Ubuntu (version 20.04.3 LTS) Client with an Nvidia RTX3090 24 GB as a graphics card with CUDA (version 11.2) for network training and evaluation. The code is written in Python (version 3.8). We used OpenSlide [42] (version 1.1.2) and Json (version 2.0.9) to access the WSIs on the fly. For preprocessing and data loading NumPy [43] (version 1.19.2), pandas [44] (version 1.2.4), pillow [45] (version 8.1.2), albumentations [46] (version 1.0.0), and OpenCV [47] (version 4.5.2) were used. For network training, we used PyTorch [48] (version 1.8.1) and progress (version 1.5) for console information. For plots matplotlib [49] (version 3.4.2), scikit-learn [50] (version 0.24.2), ptitprince [51] and seaborn [52] (version 0.11.1) were used. For further inspection of WSIs and visualization of attention maps on full WSIs, QuPath [35] (version 0.2.3) was applied.

## 7. Network Training

In each iteration, one patient with its entire set of tiles was used, and the attention part of the network was optimized using categorical cross-entropy loss and Adam [53] optimizer with a learning rate of 1 × 10^−4^ and decay of 1 × 10^−5^. In each epoch, an equal number of patients per class was used, randomly sampled from the training set. To account for class imbalances between the classes, the number of patients used per epoch is limited by the class with the lower number of patients. The network was trained for 400 epochs with early stopping after a minimum of 100 epochs (patience 25 epochs), and the best network according to validation loss is stored. Validation loss is computed on a random subset of the validation set with an equal number of patients per class per epoch.

## 8. Computational Analyses and Statistics

Computational analyses were performed in matplotlib [49], ptitprince [51], and seaborn [52] using default parameters to generate raincloud [51] and box plots. Raincloud plots show the probability density as well as raw data points overlaid with box plots. Standard box plots visualize the five-number summary (minimum, lower quartile, median, upper quartile, and maximum) of the indicated data sets. ROC (receiver (or relative) operating characteristic) curve plots were computed using scikit-learn, and the mean was computed per threshold step. In the standard ROC curve plots, sensitivity values are plotted against specificity values. Balanced accuracy (BA) values are calculated as BA = (Sensitivity + Specificity)/2.

## 9. Methylation Class Determination from WSI Data Does Not Depend on Tumor Grade

Although methylation classes share similar clinical behavior and transcriptomic profiles, different CNS WHO grades are represented to varying degrees within each class. For instance, grades 2 and 3 are overrepresented in intermediate and malignant subclasses. The uneven distribution of tumor samples with different grades within individual methylation classes and the corresponding differences in tissue structure could serve as a distinguishing feature primarily in the pairwise classification of benign versus other methylation classes. This could potentially affect AI-based histopathologic classification. In our narrow validation Dataset B, the grade 1/grade 2 distributions are 12/1 for benign-1, 17/2 for benign-2, and 6/12 for the intermediate-A cohorts (Table 1). The high diagnostic performance and their similar WHO grade distribution exclude possible grade-based class prediction in the benign-1 vs. benign-2 case. To exclude the possibility that the performance of the AI-assisted morphological diagnosis of benign-1 vs. intermediate-A methylation classes is influenced by tumor grade, we calculated and compared the diagnostic performance of sample sub-cohorts belonging to the same WHO grade. As this is infeasible for benign-1 subtypes, we compared sensitivities of intermediate-A CNS WHO grade 1 and 2 subtypes (i.e., computed the sensitivity for intermediate-A with CNS WHO grade 1 and sensitivity for intermediate-A with CNS WHO grade 2 and computed their differences per run). The result clearly showed that the sensitivity of the classification is independent of the tumor grade, as their difference for each run was closer to zero. A classification that relies on the WHO grade would result in a difference of sensitivities closer to one, with different sensitivities of the subtypes due to a misclassification of one (i.e., classifying likely most intermediate-A CNS WHO grade 1 as benign-1 and most intermediate-A CNS WHO grade 2 as intermediate-A).

## 10. Statistical Assessment of Attention Maps

We evaluated the robustness of the attention network’s focus on different areas over multiple runs of the Monte Carlo cross-validation and assessed the proportions of high-attention tiles in 20× magnification vs. 40× magnification. We were able to observe that high and low attention tiles remained particularly important across runs, implying that the focus of the network and, therefore, the attention assignment remained fairly stable across multiple runs even with changing training data. Interestingly, the network seemed to focus more on 20× magnification tiles as their high-attention proportion increased compared to 40× magnification. Information from 20× magnification includes more global tissue structures, which seemed to be more relevant for the classification. 

## Figures and Tables

**Figure 1 cancers-15-05190-f001:**
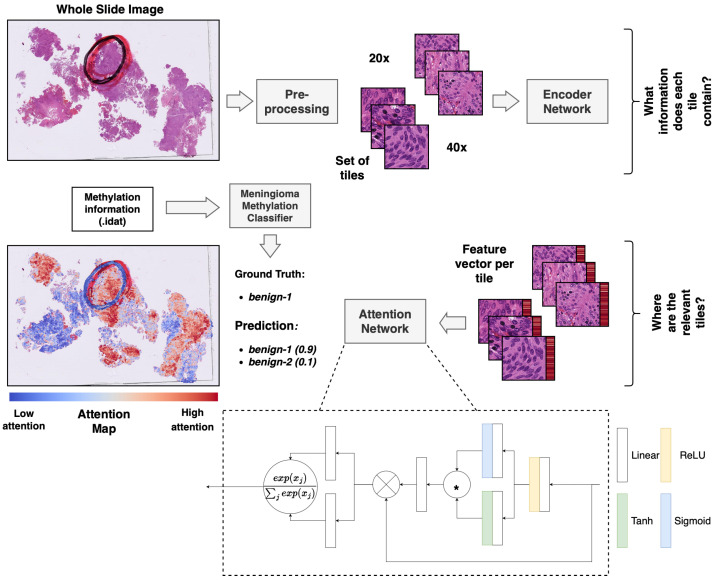
Overview of the multiple-instance deep-learning framework. Standard HE-stained histopathology slides were scanned, and the resulting WSIs were tiled. The tiled images were encoded utilizing a pre-trained ResNet50 encoder network to obtain a feature vector per image patch (red bar next to the tile). The attention network was trained using a multiple-instance approach to weigh and aggregate the patches of a WSI to make a prediction of the methylation class based on the WSI. The weights from the attention network were used to obtain an attention map, visualizing the relative importance of the tiles for the prediction. * equals elementwise multiplication.

**Figure 2 cancers-15-05190-f002:**
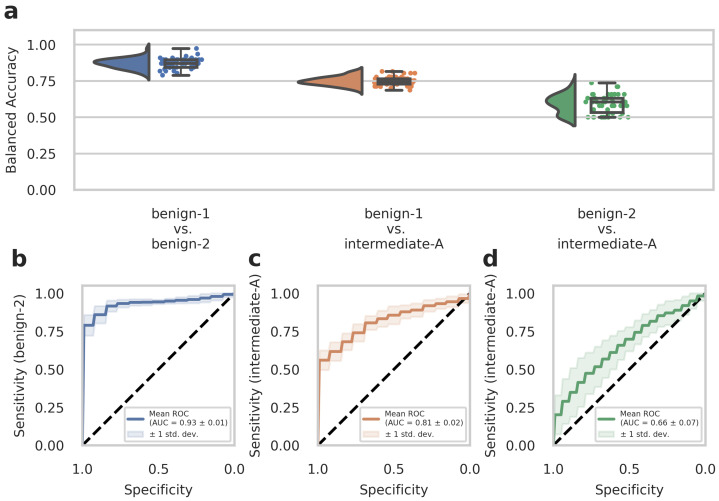
The deep neural network’s performance on the retrospective narrow validation Dataset B. (**a**) Raincloud plots (including probability distribution, box-whiskers plot, and data points from left to right) show balanced accuracy values representing discrimination power for each classification problem. Each data point represents one run of the cross-validation. (**b**–**d**) ROC curves illustrate the diagnostic performance of each pairwise classification. Visualized is the mean ROC curve over the runs. They correspond to the same pairwise combinations as the raincloud plots above. The plots show the sensitivity of one class per two-class classification problem, as labeled on the y-axis, against specificity on the x-axis. Transparent areas indicate one standard deviation of the mean curve. ROC curves represent the model’s prediction at different prediction score thresholds. A larger area under the curve represents a more confident model. Methylation classes are benign-1 (n = 13), benign-2 (n = 19), and intermediate-A (n = 19).

**Figure 3 cancers-15-05190-f003:**
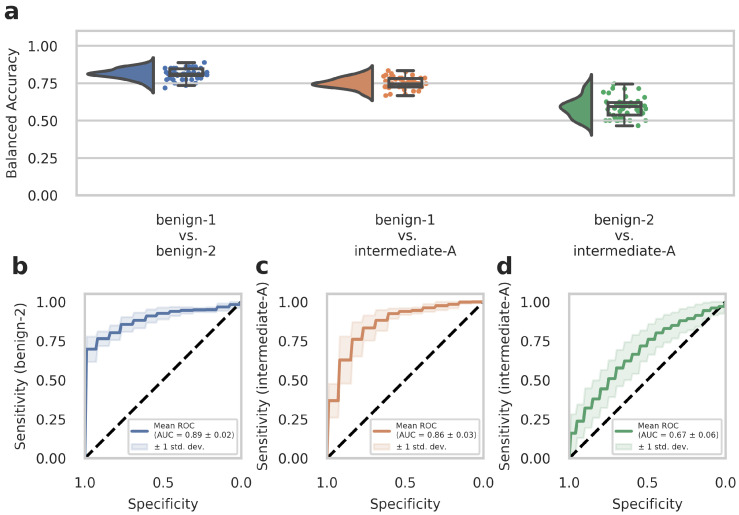
The deep neural network’s performance on Dataset D of the retrospective development Dataset. Dataset D WSIs are generated from patient samples of Dataset B, but from independent tissue sections, on which the cut-out area for methylome analysis is labeled as described in detail in the Methods section. (**a**) Raincloud plots (including probability distribution, box-whiskers plots, and data points from left to right) show balanced accuracy values representing discrimination power for each classification problem. Each data point represents one run of the cross-validation. (**b**–**d**) ROC curves illustrate the diagnostic performance of each pairwise classification. Visualized is the mean ROC curve over the runs. They correspond to the same pairwise combinations as the raincloud plots above. The plots show the sensitivity of one class per two-class classification problem, as labeled on the y-axis, against specificity on the x-axis. Transparent areas indicate one standard deviation of the mean curve. ROC curves represent the model’s prediction at different prediction score thresholds. A higher area under the curve represents a more confident model. Methylation classes are benign-1 (n = 13), benign-2 (n = 18), and intermediate-A (n = 17).

**Figure 4 cancers-15-05190-f004:**
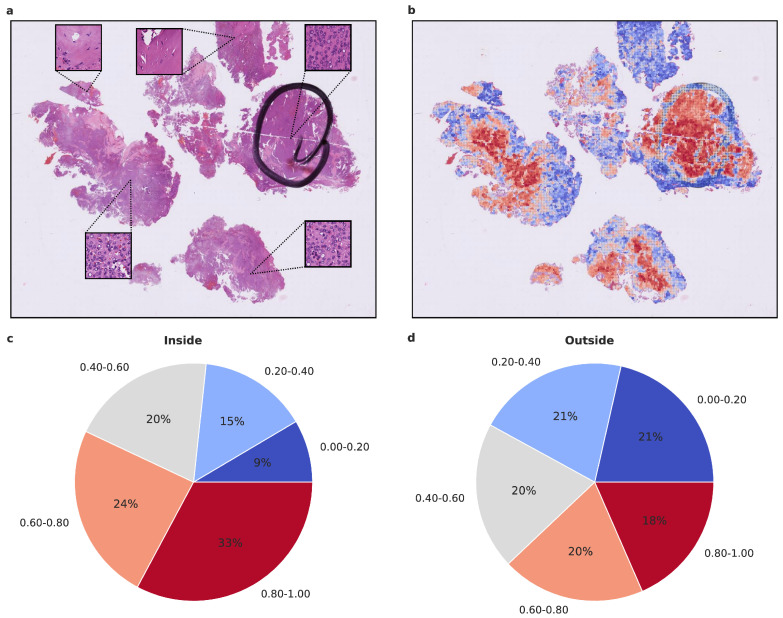
Visualized attention map and attention map statistics of benign-1 vs. benign-2 classification. (**a**,**b**) For qualitative evaluation the location and attention value as a red-blue color grade of 20× magnification tiles of WSIs is visualized (**b**) and the corresponding HE-image is shown (**a**). The attention map of a representative tumor section visualizes the relative importance of each tile for the overall prediction. A low importance is illustrated by dark blue and high importance in dark red on the color map. High-attention tiles are enriched in tumor areas, which becomes especially visible when compared to the section area preselected for DNA extraction by the neuropathologist. (**c**,**d**) Attention map statistics to evaluate the relative proportion of high and low attention values inside (**c**) and outside (**d**) the tumor areas, highlighted by a neuropathologist. The color grades were grouped into 5 discrete classes. Only attention maps at 20× were considered for this evaluation and proportion was computed per attention map and averaged overall. A higher proportion of high-attention tiles inside the marked area is in line with a high tumor area percentage inside this area. Methylation classes are benign-1 (n = 13) and benign-2 (n = 18).

**Figure 5 cancers-15-05190-f005:**
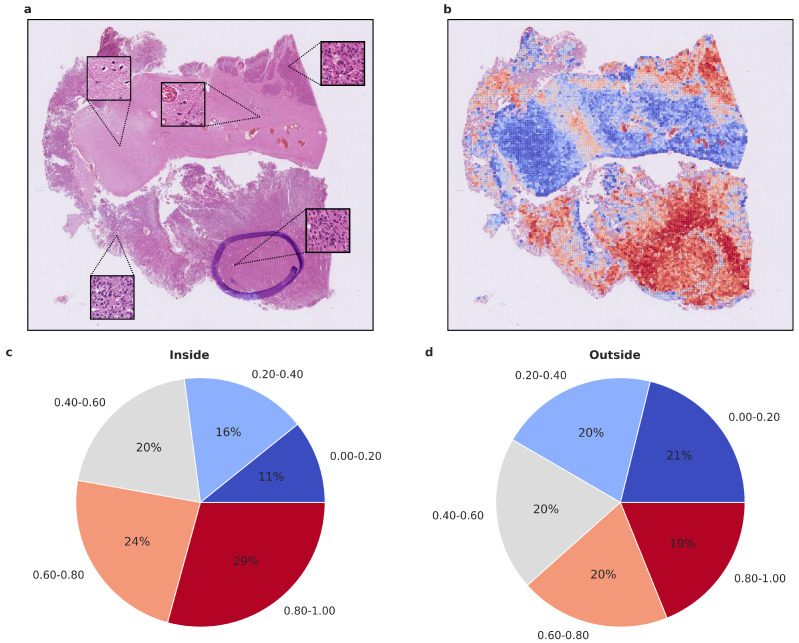
Visualized attention map and attention map statistics of benign-1 vs. intermediate-A classification. (**a**,**b**) For qualitative evaluation, the location and attention value as a red-blue color grade of 20× magnification tiles of WSIs is visualized (**b**), Colors can be interpreted as representing a certain range in percentage of attention. Blue 0–20% (least important), Light blue 20–40%, Gray 40–60%, Orange 60–80%, Red 80–100% (most important), and the corresponding HE-image is shown (**a**). The attention map of a representative tumor section visualizes the relative importance of each tile for the overall prediction. A low importance is illustrated by dark blue, and a high importance in dark red on the color map. High-attention tiles are enriched in tumor areas, which becomes especially visible when compared to the section area preselected for DNA extraction by the neuropathologist. (**c**,**d**) Attention map statistics to evaluate the relative proportion of high and low attention values inside (**c**) and outside (**d**) the tumor areas, highlighted by a neuropathologist. The color grades were grouped into 5 discrete classes. Only attention maps at 20× were considered for this evaluation, and proportion was computed per attention map and averaged overall. A higher proportion of high-attention tiles inside the marked area is in line with a high tumor area percentage inside this area. Methylation classes benign-1 (n = 13) and intermediate-A (n = 17).

**Table 1 cancers-15-05190-t001:** Overview of patient sample grouping. Numbers denote patient samples from each meningioma methylation class (according to Sahm et al. [32]) in Datasets A–D. Numbers in parentheses correspond to sample numbers with WHO grade 1/2/3. The samples used for AI-based classification are in bold.

	Group A				Group B		
	ben-1	ben-2	ben-3	int-A	int-B	malig	analyzed
A	**34**	**52**	4	**56**	6	4	**142**
	**(24/9/1)**	**(42/10/0)**	(4/0/0)	**(20/36/0)**	(1/5/0)	(0/4/0)	**(86/55/1)**
B	**13**	**19**	5	**19**	1	6	**51**
	**(12/1/0)**	**(17/2/0)**	(3/1/0)	**(6/12/0)**	(0/1/0)	(0/5/1)	**(35/15/0)**
C	**34**	**52**	4	**53**	6	4	**139**
	**(24/9/1)**	**(42/10/0)**	(4/0/0)	**(19/34/0)**	(1/5/0)	(0/4/0)	**(85/53/1)**
D	**13**	**18**	5	**17**	1	6	**48**
	**(12/1/0)**	**(16/2/0)**	(3/1/0)	**(6/10/0)**	(0/1/0)	(0/5/1)	**(34/13/0)**

## Data Availability

The code is available at github.com/Neuropathology-Giessen/METHnet, accessed on 25 October 2022.

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
