# Peer review of "Leveraging Attention-Based Convolutional Neural Networks for Meningioma Classification in Computational Histopathology"

_cancers, 2023, doi:10.3390/cancers15215190_

Round 1

Reviewer 1 Report

Comments and Suggestions for Authors

The authors developed and assessed an attention-based multiple-instance deep neural network for predicting meningioma methylation classes using tumor methylome data. The goal was to create a morphology-based classifier that would produce the same 6-tier classification as the methylome analysis. They trained the model on three groups of meningiomas found in their institute. The model demonstrated good classification performance in separating Ben-1 from Ben-2 and Ben-1 from int-A, but not Ben-2 from int-A

This well-planned study could lead to the development of a new, more accurate way to diagnose meningiomas and predict their prognosis. The authors addressed a major limitation of this study, the small cohort size for some methylation classes.

Several questions-

“no augmentation was performed on the tile level besides image normalization as needed for ResNet50”- Because the number of slides available for training is small, the authors should consider augmenting their training data

The authors applied the ResNet50 network, where the first layers were pre-trained on ImageNet, similar to Lu et al. [ref3]. However, several recent reports have shown that pre-training on pathology data outperforms supervised pre-training on ImageNet (e.g., arXiv:2212.04690). Therefore, the authors should consider pre-training their model on pathology data to see if it improves their results.

The authors speculate that the methylome proximity between Ben-2 and int-A underlies its diagnostic HE performance. Did the authors consider training a similar MIL algorithm based not on HE but on IHC used for meningioma diagnosis? (SSRT2a or EMA for example). This will highlight different areas compared to the whole HE slide/pathologist annotated regions

Author Response

Please find attached our response to the review.

Reviewer 2 Report

Comments and Suggestions for Authors

The authors present an interesting method for Leveraging Attention-Based Convolutional Neural Networks for

Meningioma Classification in Computational Histopathology. The topic is topical, and the results are well presented and structured. Also, the results are appropriate for the proposed theme.

I propose to the authors to carry out in the Discussions chapter, a comparison of the results obtained by them with the results reported in the specialized literature on the same topic. Why is their approach superior to other methods presented in the specialty literature.

Author Response

Please find attached our response to the comments.

Reviewer 3 Report

Comments and Suggestions for Authors

In summary, this study proposes an attention-based multiple-instance convolutional neural network for meningioma methylation classification in computational histopathology. The authors utilized QuPath to visualize the attention map to enable an inspection of the importance regions for the prediction, offering insights into the decision-making process of the CNN. The authors obtained a 95% confidence interval for balanced accuracy of 0.896±0.151 for the benign-1 vs. benign-2 combination and 0.830±0.146 for benign-1 vs. intermediate-A. However, the model shows to be unable to distinguish benign-2 vs. intermediate-A effectively, achieving a 95% confidence interval of 0.542±0.176. External evaluation on an independent dataset of 51 meningioma patient samples was also performed to evaluate the generality of the proposed method.

Aim

The implicit aim is to test the use of the network for meningioma classification based on labelled training data consisting of H&E-stained histological sections, with classification accuracy being the primary metric of interest.

Clarity

The work is clearly presented, and the authors state that the manuscript conforms to the recently proposed MI-CLAIM standard.

Importance

Meningiomas are a common form of intracranial tumour, and prognosis is based on tumour grading. Methylome profiling is one of a number of methods used to provide biomarkers for meningioma classification and grading (ref. 26 in original manuscript). Classification accuracy is hard to evaluate for any given method as classifications and corresponding grades are frequently updated. The study is more of a proof-of-concept that significant advance towards applying deep learning techniques to the problem being evaluated.

Ethical concerns

The authors should state in the body of the text (section 2.7) from whom informed consent was received.

Suitability of methods

In general, the methods are quite suitable for the problem being investigated. The only weak point is that two evaluations are performed, and between these, two methodological changes are made. I suggest that for data sets C and D, the authors first test the effect of using representative tumour sections, and then adding the methylation information, in order to be sure that the null results they report (compared to data sets A/B) are not the combination of one positive and one negative alteration.

Correctness of analytical procedures

There are no problems to note regarding correctness. The attention analysis is weak, because there is no comparison of attention map statistics in cases where diagnosis was correct or incorrect. The authors state that “these results indicate that the network can detect [..] regions [of interest] and preferentially aggregates its prediction from them”, but given the variable accuracy results, is this uniformly true? A breakdown of results based on the correctness of the result should be included. 

Quality of presentation

Section 2.5. The Attention Network is an important block in the proposed model. Using a framework diagram to make its design easier for readers to understand.

Figure 1. Try to represent the inputs and outputs of the two parts more clearly. Where does the methylation information enter into the scheme? “What information contains each tile?” should be “What information does each tile contain?”

Discussion of uncertainties and bias

I am not familiar with methylation classification methods. Does the restriction of the sample to patients with methylation classification probability score > 0.5 bias the sample relative to the general meningioma population?

The authors should clarify section 3.3. I suspect that the neuropathologist selective representative tumour sections in datasets (C, D) blinded to final classification. If this were not the case, then clearly the results would not represent a fair comparison.

Minor correction

Figure 2. a Raincloud plots

Author Response

Please find attached the response to the comments.

Round 2

Reviewer 1 Report

Comments and Suggestions for Authors

The authors have addressed my concerns.

They should include their explanation for the decision not to use augmentation during training in the manuscript, not just in the response letter.

Author Response

Please find the response attached.

Reviewer 3 Report

Comments and Suggestions for Authors

I thank the authors for their improved manuscript which is now quite suitable for publication.

I am happy to see that the authors have now analysed the attention maps in greater detail. In fact, such maps offer an invaluable tool for peeking inside the "black box". In this case we have learned that even when the network makes a wrong classification, at least it is looking in the right place.

Author Response

Please find attached the response.
